# Breeding of the Long-Grain Restorer of *Indica-Japonica* Hybrid Rice by Using the Genetic Effects of Grain Shape QTLs

Keke Liu [1,†], Zequn Peng [1,†], Zhihao Sun [1,†], Zhengping Zhou [1], Yanhui Li [1], Ran Zhou [1], Dengmei He [1,2,3], Chenbo Huang [1], Daibo Chen [1], Shihua Cheng [1], Liyong Cao [1,2], Xiaodeng Zhan [1,*] and Lianping Sun [1,3,*]

1   Key Laboratory for Zhejiang Super Rice Research, Chinese National Center for Rice Improvement and Stat Key Laboratory of Rice Biology, China National Rice Research Institute, Hangzhou 311402, China
2   Baoqing Northern Rice Research Center, Northern Rice Research Center of China National Rice Research Institute, Baoqing 155600, China
3   College of Agronomy, Heilongjiang Bay Agricultural University, Daqing 163711, China
*   Correspondence: zhanxiaodeng@caas.cn (X.Z.); sunlianping@caas.cn (L.S.); Tel.: +86-571-6337-0265 (X.Z.); +86-571-6337-0338 (L.S.)
†   These authors contributed equally to this work.

**Abstract:** Grain shape improvement, which determines grain yield, quality traits and commercial value, is an extremely important aspect of rice breeding. Grain size is controlled by multiple genes, and Maker Assistant Selection (MAS) breeding is effective for breeders in developing stable and efficient markers to aggregate these genes in order to speed up the selection of new lines with desirable traits during the breeding process. In this study, functional markers were developed based on the sequence differences of five grain-shaped genes (*GL7*, *GW6a*, *GS6*, *GW5* and *TGW6*) between the long-grain *japonica* rice variety Zhendao and the *indica-japonica* restorer R2027. We then constructed a population of recombinant inbred lines (RILs) based on their cross. The newly designed functional markers were used to genotype grain-size genes, and a genetic effect analysis was conducted to screen high-quality long-grain restorers. Our results reveal diverse effects of different genes on grain size, and the five genotypes were distributed in the 36 selected $BC_1F_8$ lines. Specifically, *gw5* positively regulates grain width and 1000-grain weight, *gl7* and *gs6* positively regulate grain length but negatively regulate grain width and 1000-grain weight, *tgw6* positively regulates grain length and *gw6a* positively regulates 1000-grain weight. The most outstanding outcome is that 5 of the 36 lines achieved in this study showing an excellent performance of long grain and yield characters are ideal materials not only for studying the interaction and genetic effects between polygenes but also as restorers or donors for dominant genes in *indica-japonica* hybrid rice breeding.

**Keywords:** rice (*Oryza sativa* L.); grain morphology; functional marker; quantitative trait loci; long-grain restorer





## 1. Introduction

Rice is one of the most important crops worldwide, and rice production has doubled in most regions over the past 50 years. However, in the face of the challenges of a deteriorating global environment, a growing population and a rapidly shrinking arable area, the problem of substantially increasing rice grain yields must be solved [1,2]. The grain yield of rice is mainly determined by three factors: the number of panicles per plant, the number of grains per panicle and the grain weight [3,4]. Grain size is a key determinant of rice grain appearance quality, and grain weight, which is determined by grain length, width and aspect ratio, is an important agronomic trait in artificial selection during rice breeding. Therefore, cultivating rice varieties with different grain sizes is of great significance to meeting the needs of breeders [5].

Grain shape is controlled by multiple quantitative trait loci, and many genes and QTLs related to grain shape have been cloned in rice. At least 20 genes that regulate grain shape

have been cloned, including *GW2*, *GS2*, *GS3*, *GS5*, *GW5*, *GS6*, *GW6a*, *GL7* and *GW8* [6–14]. *GL7* is a major QTL controlling the grain length on chromosome 7, which encodes a homologous protein of the *Arabidopsis thaliana* LONGIFOLIA protein. The tandem repeat upstream of this locus can up-regulate the expression level of *GL7* and down-regulate the expression level of adjacent negative regulators, regulating cell elongation and thereby increasing rice grain length and improving rice appearance quality [13,15]. An $F_2$ population was constructed by crossing the *indica* rice variety Kasalath and the *japonica* rice variety Nipponbare, and a major QTL on chromosome 5 controlling grain width, *GW5*, was located in the population. This gene encodes a calmodulin-binding protein with a 1212 bp deletion in its upstream 5 kb region. The 1212 bp deletion in wide-grain varieties regulates grain size by decreasing the expression of *GW5*. It can also inhibit the autophosphorylation of GSK2, thereby affecting the accumulation of unphosphorylated OsBZR1 and DLT proteins in the nucleus, which regulates the expression level and growth response of brassinolide-responsive genes, thereby regulating rice grain width and weight [10,16–18]. *GW6a* is a GNAT-like protein encoding histone acetyltransferase, which is localized in the nucleus and acts as a positive regulator through its upstream 1.2 k region, which is of great significance in regulating rice grain weight, yield and plant biomass [12]. As a unique member of the GRAS gene family, *GS6* is located on chromosome 6 and can negatively regulate rice grain size by decreasing the number of cells in glumes and affecting the development of palea. While *GS6* haplotypes were identified in natural materials, the sequence polymorphism of its promoter is likely to determine the grain size due to its effect on gene expression [11,19]. *TGW6* encodes a protein with IAA-glucose hydrolase activity. The large grain allele *tgw6* of Nipponbare can affect the transition from syncytium to cellularization by controlling the IAA supply and limiting the cell number. The loss of function of the *tgw6* allele in Kasalath increases the grain weight through pleiotropic effects on the source organ, thereby increasing the rice yield [20].

The expression of some grain-size genes depends on the interaction of effectors to form regulatory pathways, a phenomenon found in both *GW6a* and *GL7/GW7* [21]. *GW8*, which regulates grain width in rice, can directly bind to the promoter of *GW7* and inhibit its expression. The interaction between GW7 and GW8 can result in high-yield and high-quality rice [15]. Some *japonica* rice arieties with a better grain appearance can be bred by introgressing the *gs9* mutation without reducing the grain yield, and the grain yield can be further improved by aggregating the *gs3* allele, indicating a higher grain yield and better grain quality [22]. Xie et al. constructed a recombinant inbred line using Zhenshan97 with the loss-of-function *gs3* and Nanyangzhan of the *GS3* locus and found aggregated *gs3* and *qtgw3* in addition to localizing the *qTGW3* regulating grain size. The grain length involving these two large alleles was significantly increased compared with the genotypes with each allele, implying that there was an epistatic interaction effect between the two loci [23–25].

In this study, we first analyzed the genetic background of the two parents with different rice grain-size characteristics and then explored the relationship between rice grain-size genes. Newly developed functional markers of grain-shape genes were designed for detecting and studying the aggregate regulatory effects of these genes. Our results provide theoretical reference and detection tools for the directional improvement of grain shape during the rice breeding process and the realization of new varieties with both a high yield and high quality.

## 2. Materials and Methods

### 2.1. Plant Materials

The long-grain high-quality *indica* rice Zhendao has a better plant type, more tillers and slender grains, while R2027 is an *indica-japonica* restorer line with a loose plant type, fewer tillers and small round grains. The two parents were provided by the Super Rice Breeding Research Group of the China National Rice Research Institute.

## 2.2. Validation of Grain Size Alleles Variations

Sequencing was performed based on the reported partial functional sites of the grain size regulatory genes *GL7*, *GW6a*, *GS6* and *TGW6* to determine the sequence variation, including the coding region of *TGW6*, the promoter region of *GW6a*, the 5'UTR region of *GL7*, the promoter regions of *GW5* and *GS6* in Zhendao and R2027 [26]. The sequencing results (PCR products sequencing was completed by TsingkeBiotechnology Co., Ltd. (TsingkeBiotechnology Co., Ltd., Hangzhou, Zhejiang, China)) were analyzed using SeqMan 12.3.1.48 software. A pair of insertion/deletion primers were designed to amplify and sequence the large fragment deletion in the *GW5* promoter region and were detected by 1% agarose gel electrophoresis.

## 2.3. Development of the Grain Size Allele-Specific Marker

Using Primer Premier 6 software (had.netlify.app/primer-premier-6-for-mac.html, Premier Biosoft Co., Ltd., Canada), functional InDel markers were designed on the genetic polymorphism in *GL7*, *GW6a*, *GS6* and *GW5* between the two parents. Newly designed markers were detected using 8% polypropylene gel electrophoresis and 1% agarose gel electrophoresis. The dCAPS Finder 2.0 online design tool (http://helix.wustl.edu/dcaps/dcaps.html accessed on 26 December 2022) was used to design the dCAPS primers for the *TGW6* SNP site, and the corresponding restriction enzyme was used to digest the amplified product. All the primers used in this study were shown in Table 1.

**Table 1.** Primer information on the seven functional markers targeting five grain-size genes.

| Primer Names | Primer Sequence (5′-3′) | Enzyme Site | Product Size (bp) |
|---|---|---|---|
| GW6a_InDel-1F | GACTTATCAGCCGCACTG | | 206/200 |
| GW6a_InDel-1R | CTCTTGACCCACCTTGAATA | | |
| GW6a_InDel-2F | ATGTTCGTTCTGGTCTTGA | | 216/191 |
| GW6a_InDel-2R | GCTGCCAATTCACATTACT | | |
| GS6_InDel-F | GCGATGGAGATGGAGATG | | 149/161 |
| GS6_InDel-R | AGAGTGAGAGCAGAGACC | | |
| GW5_InDel-F | GGACTAATTACAGCGATAACC | | 1627/415 |
| GW5_InDel-F | GAACGGCAGAATGAGGAG | | |
| GL7_InDel-1F | CTCACGCACATCCAACTG | | 106/117 |
| GL7_InDel-1R | ATACCACATCTCATCTCAC | | |
| GL7_InDel-2F | GTGAGATGAGATGTGGTAT | | |
| GL7_InDel-2R | TGAAATAAGCGGGAGGGA | *Sac* I | 134/116 |
| TGW6_SNP-F | CCGATAGCAGCATGAACTA | | 167/137 |
| TGW6_SNP-R | GGTCAATGCAACGATCAGAT | | |

## 2.4. DNA Extraction and PCR

Young leaves were collected at the rice tillering stage (about 30 days after sowing), and 100 mg of leaves at 3–5 cm from the tip of the flag leaf collected in the field was added to fully grind with liquid nitrogen. The improved CTAB method was used to extract genomic DNA [27]. A 20 μL system with 2 μL of template DNA, 10 μL of 2× Rapid Taq Master Mix ((TOYOBO, Osaka, Japan)), 1 μL of each primer (10 μM) and 6 μL of distilled water was subjected to PCR for most of the markers. For GW5-InDel labeling, the best amplification effect was achieved by using 25 μL of KOD FX DNA Polymerase premix (TOYOBO, Osaka, Japan) and adding 2 μL of template DNA, 1 μL (10 μM) of each primer, 13 μL of 2× PCR Buffer, 5 μL dNTPs (2 mM) and 3 μL of distilled water. The PCR program was set as follows: initial denaturation at 95 °C for 3 min, 35 cycles of denaturation at 95 °C for 15 s, annealing at the proper temperature for 20 s, extension at 72 °C for 1 min per kb and a final extension at 72 °C for 5 min. For TGW6-SNP labeling, 10 μL PCR products were digested with 2 μL 10× reaction buffer, 0.4 μL restriction endonuclease (Thermo Fisher Scientific Waltham, MA, USA) and 7.6 μL distilled water, resulting in a 20 μL system. After 1 h of digestion at 37 °C, 10 μL of the product was used for electrophoresis. GW5-InDel was

labeled by 2% agarose gel electrophoresis, and the other labels were electrophoresed by 8% polyacrylamide gel.

### 2.5. Examination of Grain Size-Related Traits

Parents and hybrid offspring groups were planted in the natural experimental plot of the China National Rice Research Institute in Fuyang, Hangzhou and Lingshui, Hainan. Two protective rows were set up around the test materials, all the materials were planted with single seedlings and the field cultivation management was the same as that of the general field. The grains on the main panicle were harvested at rice maturity and dried at 45 °C for two days. A total of 100 mature and full grains were randomly selected and placed on the i800 ScanMaker Plus scanner (Zhongjing Tech. Co., Ltd., Shanghai, China), and the grain number, grain length, grain width and aspect ratio were automatically calculated. The thousand-grain weight of the samples was calculated by applying the formula—(grain weight/grain number)*1000—three times and averaging the results. The collection, arrangement and analysis of the test data use the GraphPad Prism 8.0.2.263 software (GraphPad Software Co., Ltd., San Diego, CA, USA).

## 3. Results

### 3.1. Phenotype of Grain Size in Zhendao and R2027

The investigation of grain size traits between Zhendao and R2027 found that the grain shape of Zhendao was slenderer and longer (the grains of Zhendao were $10.20 \pm 0.35$ mm in length and $2.42 \pm 0.21$ mm in width), while the grain shape of R2027 was shorter and rounder (the grains of R2027 were $8.10 \pm 0.22$ mm in length and $2.97 \pm 0.16$ mm in width) (Figure 1A–C). The grain aspect ratio of Zhendao was up to $4.39 \pm 0.02$, which was significantly larger than that of R2027 ($2.76 \pm 0.01$), but the difference in the thousand-grain weight was not obvious (Figure 1D,E). Therefore, the selected two parents, Zhendao and R2027, were a suitable donor parent and reincarnation parent, as they showed a similar grain weight but significantly different length–width ratios. Zhendao is an ideal long-grain donor parent.

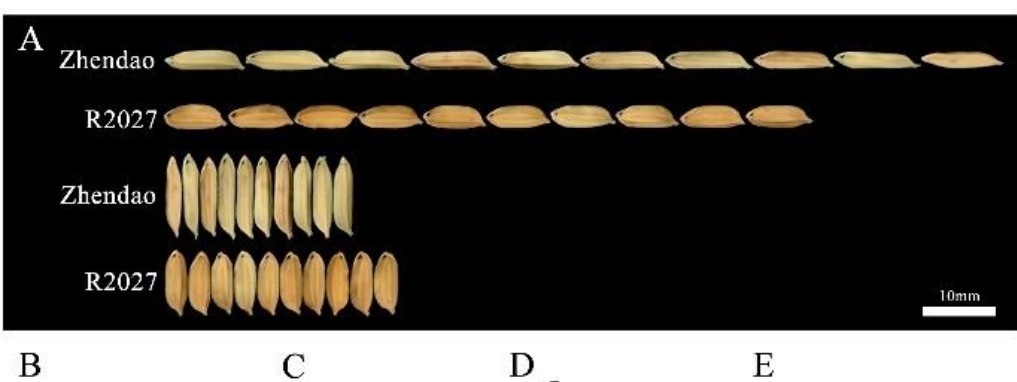

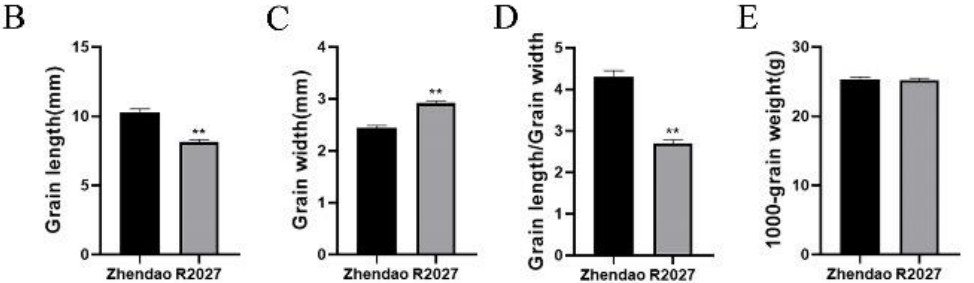

**Figure 1.** Grain size comparison between Zhendao and R2027. (**A**) Length and width comparison of Zhendao and R2027. Scale bar, 10 mm. (**B**–**E**) Grain length (**A**), grain width (**B**), grain length/grain width (**D**) and 1000-grain weight (**E**) of Zhendao and R2027. Error bars, SD of ten biological replicates. ** $p < 0.01$ compared with WT (Zhendao), Student's *t*-test.

### 3.2. Study of the Candidate Functional Variants of Five Grain-Size Genes

We sequenced nine major grain-size genes (*GW2*, *GS3*, *TGW3*, *GW5*, *GS6*, *GW6a*, *TGW6*, *GL7*, *GW8*) of Zhendao and R2027, but no differences in the *GW8*, *TGW3*, *GW2* or *GS3* genes were found between the two parents. Subsequently, we verified the polymorphism sequences of these genes using the Nipponbare genome sequence as a reference, including six InDels sites and one SNP site. The results of the *GL7* gene detection showed that an insertion/deletion mutation (18 bp insertion and 11 bp deletion) occurred in the 5′UTR region of Zhendao, which is consistent with previous findings [26] (Figure 2). The changes in these two sequence polymorphisms may affect the grain size. We also found two deletion sequences in the promoter of Zhendao *GW6a*, which are consistent with the sequence polymorphisms reported at positions −395 to −1415 of its promoter and which we consider to be one of the reasons for the variation of grain traits (Figure 2). Compared with Zhendao, a 12 bp insertion in the amplified *GS6* promoter sequence may lead to altered gene expression and affect the grain shape (Figure 2). We selected a G-to-T SNP at position 600 of the cDNA sequence of *TGW6* as a candidate functional variant which has been identified in previous research (Figure 2). The 4 kp region upstream of the *GW5* of the Zhendao and R2027 genes was amplified, sequenced and subjected to 2% agarose gel electrophoresis (Figure 2). A 1212 bp deletion was found in R2027, consistent with studies on Nipponbare.

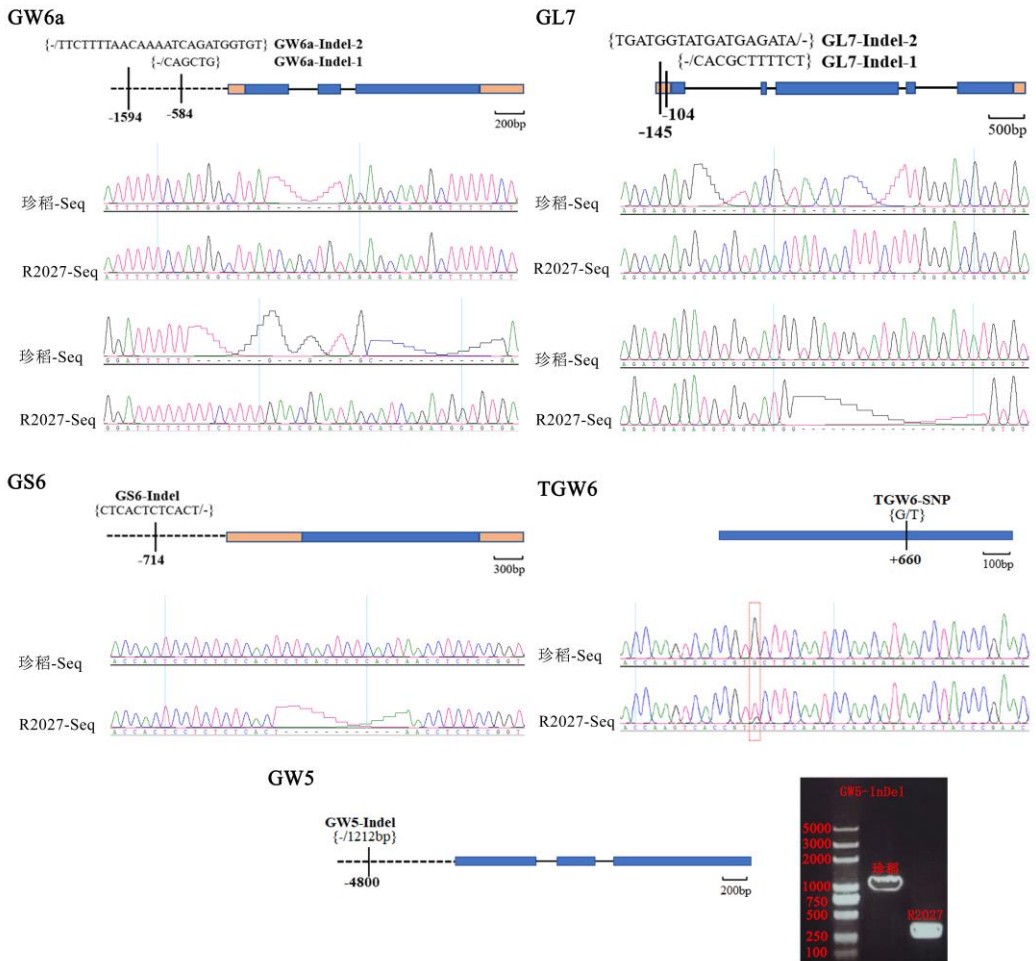

**Figure 2.** Position of the candidate SNP or InDel of five grain-size genes. The promoters, UTRs, exons and introns of the different genes are expressed as dashed lines, orange boxes, blue boxes and straight lines, respectively. The position of each variant is marked as the relative distance to ATG in the promoter (−) and CDS (+).

### 3.3. Functional Marker Development of Five Size Genes

To facilitate the examination of genotyping, we designed InDel molecular markers for *GL7*, *GW6a*, *GS6* and *GW5* and a dCAPS marker for *TGW6* based on the above sequence analysis results (Table 1). The products amplified by the GL7_InDel1 primers produce two bands of 117 bp and 106 bp, and the products amplified by the GL7_InDel2 primers will produce two bands of 134 bp and 116 bp. Of these, 117 bp and 134 bp are 5′UTR insertions of type *gl7*, while 106 bp and 116 bp are the promoter 5′UTR deletion type *GL7*. The products amplified by GW6a_InDel1 primers will produce two band types of 206 bp and 200 bp, and the amplified products of GL7_InDel2 primers produce two bands of 216 bp and 191 bp, of which 206 bp and 216 bp are the promoter insertion type *GW6a*, while 200 bp and 191 bp are the promoter-deleted type *gw6a*. For the deletion of 12 bp in the *GS6* promoter, 161 bp and 149 bp bands were generated after primer amplification, of which 161 bp was the promoter-inserted type *gs6*, and 149 bp was the promoter-deleted type *GS6*. Due to the deletion of the 1212 large fragments upstream of *GW5*, we designed the molecular marker primer GW5_InDel and identified it using 2% agarose gel electrophoresis. We found R2027 to be a 1212 bp deletion type *gw5*, and Zhendao was a non-deletion type *GW5*. The digestion products amplified by the TGW6_SNP primer showed two bands of 167 bp and 137 bp, of which 137 bp is the rare allelic variant *tgw6* type.

### 3.4. Detection of Restorer Genes in the Zhendao and R2027 Recombinant Inbred Lines

During the winter of 2017, the *indica-japonica* restorer R2027 (as the pollen donor) was crossed with Zhendao. The $F_1$ population was planted in Hangzhou during the summer of 2018, and a sequence analysis of grain-shape genes was conducted. Newly designed functional markers of *GL7*, *GW6a*, *GS6*, *GW5* and *TGW6* were filtered and determined to be available. Thirteen individual plants were identified to be heterozygous. In the winter of 2018, $F_2$ populations were planted in Hainan, 1856 individual plants were sampled and identified using the above markers and 197 individual plants with excellent agronomic characteristics and a long grain shape were retained. Further ecological identification and MAS were carried out in Hangzhou and Hainan from the $F_3$ through the $F_7$ generation (Figure 3), and 78 lines ($F_8$) were selected to be planted in Hangzhou during the summer of 2022. We further detected five grain-shape genes for genetic analysis, which we named TF1 to TF78 (TF refers to the restorer lines number of our lab during the spring of 2022 in the Lingshui Hainan breeding field).

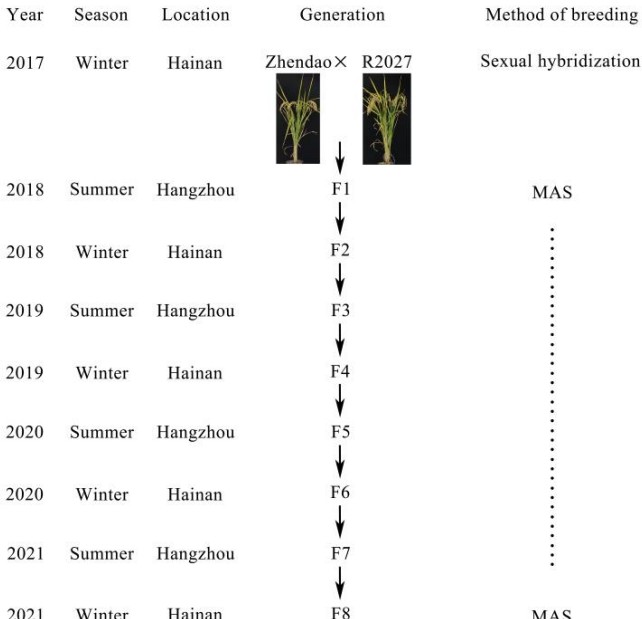

| Year | Season | Location | Generation | Method of breeding |
|------|--------|----------|------------|--------------------|
| 2017 | Winter | Hainan | Zhendao × R2027 | Sexual hybridization |
| 2018 | Summer | Hangzhou | F1 | MAS |
| 2018 | Winter | Hainan | F2 | |
| 2019 | Summer | Hangzhou | F3 | |
| 2019 | Winter | Hainan | F4 | |
| 2020 | Summer | Hangzhou | F5 | |
| 2020 | Winter | Hainan | F6 | |
| 2021 | Summer | Hangzhou | F7 | |
| 2021 | Winter | Hainan | F8 | MAS |

**Figure 3.** Breeding the hybrid progeny of Zhendao and R2027.

### 3.5. Detection of Major Restorer Genes in TF Families

　　　　The fertility restorer of male sterility in rice is the basis for related research and the utilization of three-line hybrid rice, and its genetic mechanism has long been a research hotspot. The fertility restorer genes *Rf1* and *Rf4* can restore the fertility of WA-type and BT-type CMS lines [28–31]. We further tested the major restorer gene linkage markers *Rf1* and *Rf4*. It was found that 36 restorer lines exhibited both the *Rf1* insertion type and *Rf4* insertion type in 78 TF families (Figure 4). These 36 materials can further be widely used in *indica* and *japonica* three-line hybrid rice.

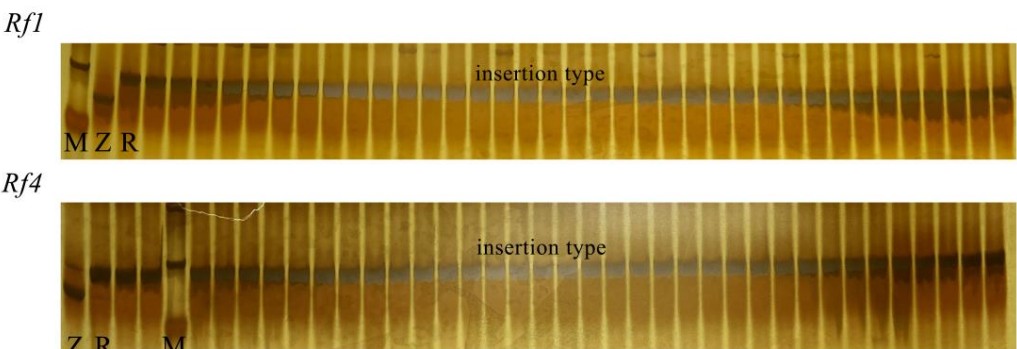

**Figure 4.** Detection of restorer genes *RF₁* and *RF₄* in 36 TF families. Z: Zhendao, R: R2027, M: DNA Marker 1.

### 3.6. Effects of the Aggregation of Different Grain-Shape Genes on the Grain Shape in TF Families

　　　　We identified the genotype of each line using the five molecular markers (Table 1). The grain length of the lines containing the *GL7* long-grain allele was 8.68–9.91 mm, significantly larger than that of the *GL7* lines containing the short-grain allele, with a grain length of 8.16–8.58 mm. The grain width of lines containing the *GW5* and *GS6* alleles was 2.51–2.66 mm, and the 1000-grain weight of the *TGW6* and *GW6a* alleles that controlled the grain weight was greater than 23.11 g (Table 2). These results demonstrated that these molecular markers can accurately detect the effects of the five different genes on grain shape and provide a basis for the breeding to improve rice grain shape. We divided the TF families into four grain-shape groups according to the grain aspect ratio: extremely elongated, slender-elongated, semi-long and general long. The grain length gene *GL7* played a major role, increasing the aspect ratio of TF1-20 above 3.5. However, the extremely long grain shape may be caused by the influence of other grain-shape genes. The grain-width genes *GS6* and *GW5* had better aggregation effects, and 15 strains had an aspect ratio of less than 3.3. *TGW6* also plays an additive role. These findings confirm the important role that these grain-shape genes play in Zhendao and R2027.

**Table 2.** Grain-type investigation and grain-size genotyping summary of 36 TF families.

| Cultivar | Grain Morphology | | | | Genotype | | | | |
|---|---|---|---|---|---|---|---|---|---|
| | GL | GW | GL/GW | TGW | *GL7* | *GW6a* | *GS6* | *GW5* | *TGW6* |
| Zhendao | 10.20 ± 0.35 | 2.42 ± 0.21 | 4.39 ± 0.02 | 25.79 ± 0.12 | *gl7* | *gw6a* | *gs6* | *GW5* | *TGW6* |
| R2027 | 8.10 ± 0.22 | 2.97 ± 0.16 | 2.76 ± 0.01 | 25.14 ± 0.34 | *GL7* | *GW6a* | *GS6* | *gw5* | *tgw6* |
| TF-1 | 9.91 ± 0.30 | 2.40 ± 0.20 | 4.16 ± 0.02 | 22.34 ± 0.25 | *gl7* | - | *gs6* | - | - |
| TF-2 | 9.87 ± 0.35 | 2.37 ± 0.14 | 4.18 ± 0.02 | 22.90 ± 0.24 | *gl7* | - | *gs6* | - | - |
| TF-3 | 9.86 ± 0.40 | 2.43 ± 0.19 | 4.07 ± 0.02 | 23.30 ± 0.27 | *gl7* | - | *gs6* | - | - |
| TF-4 | 9.73 ± 0.38 | 2.38 ± 0.21 | 4.16 ± 0.02 | 23.00 ± 0.22 | *gl7* | - | *gs6* | - | - |
| TF-5 | 9.70 ± 0.38 | 2.38 ± 0.19 | 4.10 ± 0.02 | 22.17 ± 0.29 | *gl7* | - | *gs6* | - | - |
| TF-6 | 9.64 ± 0.25 | 2.42 ± 0.20 | 4.00 ± 0.02 | 22.47 ± 0.29 | *gl7* | - | *gs6* | - | - |
| TF-7 | 9.39 ± 0.39 | 2.37 ± 0.22 | 4.00 ± 0.02 | 19.01 ± 0.16 | *gl7* | - | *gs6* | - | - |
| TF-8 | 9.39 ± 0.30 | 2.30 ± 0.17 | 4.08 ± 0.01 | 19.78 ± 0.21 | *gl7* | - | *gs6* | - | - |
| TF-9 | 9.81 ± 0.39 | 2.49 ± 0.17 | 3.95 ± 0.02 | 23.00 ± 0.15 | - | *gw6a* | *gs6* | - | - |
| TF-10 | 9.46 ± 0.39 | 2.40 ± 0.19 | 3.95 ± 0.02 | 22.32 ± 0.18 | - | *gw6a* | *gs6* | - | - |
| TF-11 | 9.42 ± 0.32 | 2.43 ± 0.18 | 3.90 ± 0.02 | 22.56 ± 0.24 | - | *gw6a* | *gs6* | - | - |

**Table 2.** *Cont.*

| Cultivar | Grain Morphology | | | | Genetype | | | | |
|---|---|---|---|---|---|---|---|---|---|
| | GL | GW | GL/GW | TGW | GL7 | GW6a | GS6 | GW5 | TGW6 |
| TF-12 | 9.37 ± 0.36 | 2.40 ± 0.26 | 3.93 ± 0.01 | 21.89 ± 0.19 | - | gw6a | gs6 | - | - |
| TF-13 | 9.34 ± 0.39 | 2.39 ± 0.18 | 3.95 ± 0.02 | 21.45 ± 0.38 | - | gw6a | gs6 | - | - |
| TF-14 | 9.19 ± 0.30 | 2.32 ± 0.15 | 3.97 ± 0.02 | 19.23 ± 0.15 | - | gw6a | gs6 | - | - |
| TF-15 | 9.15 ±0.42 | 2.31 ± 0.15 | 3.98 ± 0.01 | 20.75 ± 0.22 | - | gw6a | gs6 | - | - |
| TF-16 | 9.11 ± 0.41 | 2.30 ± 0.16 | 3.98 ± 0.01 | 20.52 ± 0.30 | - | gw6a | gs6 | - | - |
| TF-17 | 9.01 ± 0.35 | 2.39 ± 0.15 | 3.79 ± 0.02 | 20.38 ± 0.22 | - | gw6a | gs6 | - | - |
| TF-18 | 8.68 ± 0.32 | 2.34 ± 0.24 | 3.73 ± 0.01 | 18.52 ± 0.14 | - | gw6a | gs6 | - | - |
| TF-19 | 8.78 ±0.31 | 2.51 ± 0.18 | 3.52 ± 0.01 | 25.18 ± 0.18 | - | - | gs6 | - | - |
| TF-20 | 8.96 ± 0.38 | 2.53 ± 0.19 | 3.53 ± 0.02 | 25.80 ± 0.31 | - | - | gs6 | - | - |
| TF-21 | 8.90 ± 0.32 | 2.57 ± 0.16 | 3.48 ± 0.02 | 24.81 ± 0.28 | - | - | gs6 | - | - |
| TF-22 | 8.85 ±0.16 | 2.56 ± 0.18 | 3.48 ± 0.02 | 25.88 ± 0.31 | - | - | gs6 | - | - |
| TF-23 | 8.96 ± 0.43 | 2.65 ± 0.19 | 3.38 ± 0.02 | 27.88 ± 0.35 | - | - | gs6 | - | - |
| TF-24 | 8.82 ± 0.51 | 2.64 ± 0.15 | 3.35 ± 0.01 | 25.70 ± 0.33 | - | - | gs6 | - | - |
| TF-25 | 8.77 ± 0.28 | 2.66 ± 0.17 | 3.30 ± 0.02 | 27.58 ± 0.22 | - | - | gs6 | - | - |
| TF-26 | 8.77 ± 0.26 | 2.56 ± 0.18 | 3.42 ± 0.01 | 27.27 ± 0.31 | - | - | gs6 | - | - |
| TF-27 | 8.68 ± 0.31 | 2.57 ± 0.18 | 3.37 ± 0.02 | 26.75 ± 0.29 | - | - | - | gw5 | - |
| TF-28 | 8.67 ± 0.21 | 2.60 ± 0.18 | 3.33 ± 0.02 | 25.14 ± 0.28 | - | - | - | gw5 | - |
| TF-29 | 8.64 ± 0.24 | 2.55 ± 0.19 | 3.40 ± 0.01 | 23.95 ± 0.20 | - | - | - | gw5 | - |
| TF-30 | 8.58 ± 0.40 | 2.58 ± 0.17 | 3.34 ± 0.03 | 23.87 ± 0.24 | - | - | - | gw5 | - |
| TF-31 | 8.58 ± 0.20 | 2.60 ± 0.18 | 3.31 ± 0.01 | 23.18 ± 0.18 | - | - | - | gw5 | tgw6 |
| TF-32 | 8.53 ± 0.28 | 2.58 ± 0.16 | 3.30 ± 0.01 | 24.75 ± 0.32 | - | - | - | gw5 | tgw6 |
| TF-33 | 8.49 ± 0.28 | 2.63 ± 0.19 | 3.24 ± 0.02 | 24.30 ± 0.29 | - | - | - | gw5 | tgw6 |
| TF-34 | 8.46 ± 0.31 | 2.59 ± 0.16 | 3.28 ± 0.00 | 23.91 ± 0.31 | - | - | - | - | tgw6 |
| TF-35 | 8.27 ± 0.32 | 2.61 ± 0.16 | 3.17 ± 0.01 | 23.11 ± 0.18 | - | - | - | - | tgw6 |
| TF-36 | 8.16 ± 0.21 | 2.62 ± 0.21 | 3.12 ± 0.01 | 23.72 ± 0.20 | - | - | - | - | tgw6 |

After agronomic character inspection (Supplementary Table S1), we finally selected five high-quality lines as potential restore resources for future breeding (Figure 5A). These five lines exhibited comparatively ideal agronomic traits such as an appropriate plant height, multi-tillers and an improved yield per plant (Table 3). Above all, the grain length of these five lines was significantly longer than that of R2027 (Figure 5B). These selected lines may provide excellent restorer resources with potential breeding value for subsequent three-line hybrid rice breeding.

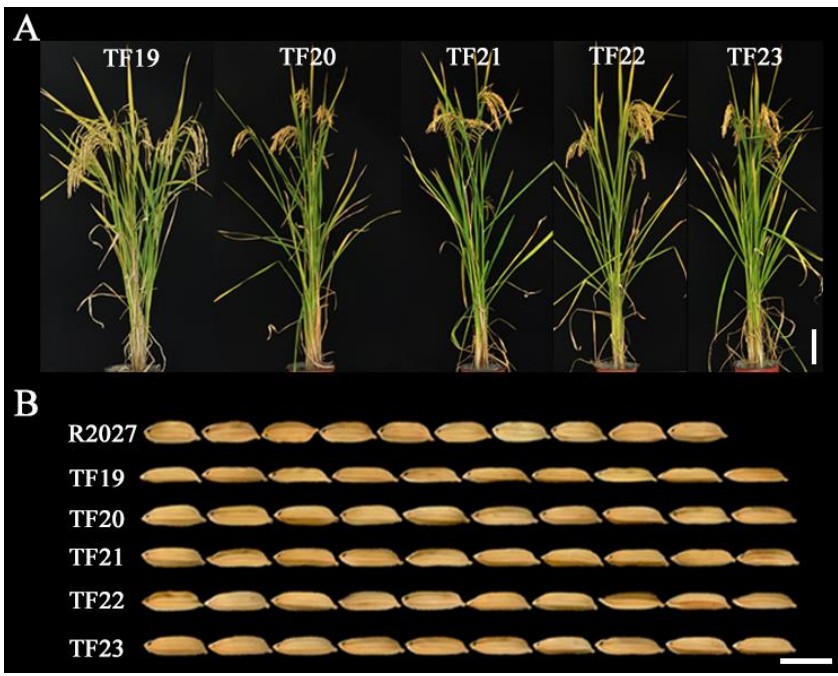

**Figure 5.** Phenotypes of five TF lineage plants. Scale bar = 10 cm for (**A**) and 1 cm for (**B**).

**Table 3.** Comparison of the main agronomic traits between five TF lineage plants and two parents (mean ± SD, n = 10).

| Trait | Zhendao | R2027 | TF19 | TF20 | TF21 | TF22 | TF23 |
|---|---|---|---|---|---|---|---|
| Plant height (cm) | 132.11 ± 1.02 | 134.26 ± 1.32 | 125.08 ± 1.32 | 132.66 ± 0.68 | 130.70 ± 1.21 | 130.72 ± 0.69 | 128.72 ± 0.69 |
| Tiller number | 6.20 ± 1.50 | 5.80 ± 1.40 | 11.80 ± 1.80 | 6.00 ± 1.20 | 7.00 ± 1.60 | 6.40 ± 1.20 | 6.80 ± 1.30 |
| Yield per plant (g) | 25.23 ± 4.11 | 26.01 ± 3.77 | 27.75 ± 2.73 | 26.66 ± 3.80 | 27.66 ± 3.10 | 27.14 ± 2.55 | 26.44 ± 3.08 |
| 1000-grain weight (g) | 25.79 ± 0.12 | 25.14 ± 0.34 | 25.18 ± 0.18 | 25.80 ± 0.31 | 24.81 ± 0.28 | 25.88 ± 0.31 | 27.88 ± 0.35 |
| Seed-setting rate (%) | 79.82 ± 0.89 | 70.82 ± 0.68 | 83.40 ± 0.45 | 81.70 ± 0.69 | 79.20 ± 1.02 | 80.20 ± 0.79 | 81.10 ± 0.43 |

## 4. Discussion

Grain length affects not only yield traits but also the appearance quality and commercial value of rice. Therefore, most breeders and breeding units are highly concerned with the improvement of rice grain shape. Molecular markers of related genes play a very important role in the implementation of MAS during the breeding process to improve grain shape [32,33]. Beneficial alleles of *GS3* and *Gn1a* have recently been introduced into the high-quality cultivar Kongyu131, which is widely grown in Northeast China, improving the grain yield and regional adaptability of the newly developed lines [34]. Many gene-linked markers have been developed to facilitate MAS breeding, but they are not associated with the target allele of the gene and can only be used for allele mining after cloning the target allele [35]. In contrast, functional markers developed from pathogenic variants are diagnostic of desired trait alleles, and the breeding efficiency of resistant materials has been greatly improved with the help of MAS [36,37].

To apply MAS breeding better, we developed a wider range of functional markers covering the major genes for grain shape, demonstrating the role of these markers in the identification of grain size traits [26]. With the availability of these functional markers, grain size can be engineered by introgressing or aggregating alleles as part of the breeding process. In the past two years, thirty-five granular QTLs have been detected on 11 chromosomes. The main cluster of QTL, *qGS7*, which was consistent with *GL7/GW7*, was cloned and showed strong effects on both grain length and grain width [38]. Zhao et al. found that the introduction of the *gs9* allele into elite rice varieties could significantly improve grain morphology and appearance quality [39]. Long, slender and semi-long grain varieties may all be targeted, depending on the needs of the breeders and the breeding objectives. According to their grain-shape gene contributions, the GW5_InDel, GL7_InDel1 and TGW6_SNP alleles can be used to produce short and thick rice, and their contrasting alleles can be used to produce slender rice. By examining the grain-shape-related traits of the TF families that aggregated different genotypes, it was found that the TF families that aggregated *gw5* and *tgw6* had the best overall performance in yield-related traits, followed by the families containing *gw5* and *tgw6*, respectively. According to the QTL analysis by Wang et al. [14], some loci can only be detected in populations without *GW5*; hence, we speculate that different *GW5* alleles show different contributions combined with other loci. The TF lineages that aggregated *gl7* and *gs6* played an important role in controlling grain length but reduced the thousand-grain weight. It shows that, to achieve the high-quality and high-yield goal of breeding new rice varieties at the same time, it is not just a simple aggregation of multiple dominant genes but also a comprehensive investigation of multiple traits and a reasonable combination of dominant genes [13,15,19].

To achieve MAS, it is first necessary to confirm the class of alleles missing in the receptor using the marker set developed in this study. In the southern US rice germplasm, it was also important to establish a set of routine genomic selection markers for the genetic improvement of variety adaptability and demonstrate their validity [40]. Based on the above results, it is necessary to determine which grain-shape genes can be aggregated to obtain the desired grain shape. Currently, InDel polymorphisms can be easily detected based on polymerase chain reaction (PCR) fragment length polymorphisms. Most of the markers we developed are InDel markers, which can be easily transferred to molecular markers and can implement most breeding programs with reasonable equipment expenditure, among

the different DNA polymorphisms [41,42]. PCR-gel-based SNP markers can be developed by an allele-specific PCR method or the CAPS/dCAPS method, but these methods have the limitation of non-specific amplification, they are only effective for specific SNP types and they have a lower chance of primer synthesis success [43]. InDel markers have naturally become a priority for breeders in genotyping large amounts of material and simultaneously analyzing the effects of different genes. Since the 1970s, the BT-type and WA-type CMS lines have been widely used in *indica* and *indica-japonica* three-line hybrid rice breeding and have also become the most extensive representative of three-line hybrid rice. So far, the hybrid seed production of many cultivated crop varieties has been performed using a three-line hybrid breeding system [44,45]. It had been confirmed that the *Rf4* gene restores fertility through the sporophyte mechanism, but *Rf4* pollen grains show preferential fertilization in experimental $F_1$ hybrid plants [46]. After agronomic characterization, five high-quality lines exhibiting a more suitable plant type, slender grains and a higher yield and carrying both of the two restorer genes were finally selected. These materials can be used as potential restorer resources for three-line hybrid rice breeding, including *indica*, *indica-japonica* and *japonica* hybrid rice types.

In this study, corresponding functional markers were designed according to the sequencing results of five grain-shape genes of the parents Zhendao and R2027, and the markers were identified using the recombinant inbred progeny lines. We found that *tgw6*, *gw5* and *gs6* in some TF families exist alone, and we detected QTL allelic variation sites such as *GW8*, *TGW3*, *GW2* and *GS3*, but none of them showed polymorphisms. This indicated that the interaction effect of these grain-shape genes plays an important role in controlling grain length, grain width and thousand-grain weight. However, only two genes were aggregated, and the genetic mechanism requires further exploration.

In summary, our research further confirmed the wide range of genetic diversity of the rice grain shape between different resources. The accurate analysis of the grain-shape genes of the specific rice germplasm and their genetic effects, as well as the functional markers and gene aggregation of grain-shape genes, can provide more reliable theoretical support and valuable breeding resources for rice breeding.

## 5. Conclusions

In the current study, we developed functional markers for five grain-size genes which were found to have corresponding genotypes in the two pedigrees, elucidating the ability of different alleles. We compared the grain-shape traits by genotyping these genes in the pedigree, finding that five markers were significantly associated with the grain shape traits and a different combination of different grain-shape genes has significant genetic diversity. The associated grain-shape marker set can provide association and linkage analysis for future genetic studies and can be an effective tool for the rational design of rice grain size. We further detected the restorer genes *Rf1* and *Rf4* in the recombinant inbred lines and found 36 lines in total; 5 lines showing excellent performance were finally selected as potential restorer resources for three-line hybrid rice breeding.

**Supplementary Materials:** The following supporting information can be downloaded at: https://www.mdpi.com/article/10.3390/agronomy13010107/s1, Table S1: Agronomic traits of 31 restorer lines achieved in this study.

**Author Contributions:** L.S., X.Z. and K.L. designed and conceived the study. K.L., Y.L., Z.S., R.Z., C.H. and D.H. performed the experiments. K.L., Z.P. and Z.S. analyzed the experimental data. Z.Z., Z.P. and D.C. completed the field trial work. S.C. and L.C. provided funding and supervised the work. K.L. wrote the first draft. L.S., X.Z. and K.L. completed the writing and reviewed the manuscript. All authors have read and agreed to the published version of the manuscript.

**Funding:** This research was supported by funding from the China Agriculture Research System, grant number (CARS-01-03); the Zhejiang Provincial Key Special Projects, grant number (2021C02063-1); the Project in Command of Hainan Yazhou Bay Seed Laboratory, grant number B21HJ0219; the Natural

Science Foundation of Zhejiang Province of China, grant number (LY21C130003); and the Chinese Academy of Agricultural Sciences Innovation Project, grant number CAAS-ASTIP-2013-CNRRI.

**Data Availability Statement:** Not applicable.

**Conflicts of Interest:** The authors declare no conflict of interest.

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
