# Peer review of "Breeding of the Long-Grain Restorer of Indica-Japonica Hybrid Rice by Using the Genetic Effects of Grain Shape QTLs"

_agronomy, doi:10.3390/agronomy13010107_

Round 1

Reviewer 1 Report (Previous Reviewer 4)

authors have done all the requireds 

Author Response

Dear Reviewer:

  Thank you very much for your careful review and important comments. We have further improved the manuscript and made some changes in the manuscript. These changes will not influence the content and framework of the manuscript.

Reviewer 2 Report (Previous Reviewer 3)

The manuscript “Breeding of long-grain restorer of indica-japonica hybrid rice 3 by using genetic effects of grain shape genes” is well written and the results can help the realization of new rice varieties with both high yield and high quality.

-        You should improve the formatting of Table 1

-        Delete the extra space in the line 233

Author Response

Dear Reviewer

  Thank you very much for your careful review and valuable suggestions. We have modified the format of Table 1 and removed the extra space in the line 233. We have further improved the manuscript and made some changes in the manuscript. These changes will not influence the content and framework of the manuscript.

Reviewer 3 Report (Previous Reviewer 1)

The authors have address the questions listed in my last review and the manuscript has been properly improved. I don't have any question about it.

Author Response

Dear Reviewer:

  Thank you very much for your careful review and important comments.  We have further improved the manuscript and made some revisions to the manuscript.  These changes do not affect the content and framework of the manuscript.

This manuscript is a resubmission of an earlier submission. The following is a list of the peer review reports and author responses from that submission.

Round 1

Reviewer 1 Report

The manuscript titled "Breeding of long-grain restorer of indica-japonica hybrid rice by  using genetic effects of grain shape genes and newly developed  functional markers" provides some interesting results and the conclusion is significant for rice breeding. The manuscript was well written and results were presented logically. Here are some minor comments about this ms.

1. The title is not consice enough, please revise it.

2. Data analysis and statistical analysis should be described in methods section.

3. What is the meaning of error bar? please describe in the figure legend.

4. The discussion part can be improved by further comparing the relationship of those five grain shape related genes, including any possible crosstalk among them.

Author Response

Dear Editors and Reviewers:

  Thank you for your letter and for the reviewers’ comments concerning our manuscript entitled “Breeding of long-grain restorer of indica-japonica hybrid rice by using genetic effects of grain shape genes” (ID: 1972460). Those comments are all valuable and very helpful for revising and improving our manuscript, as well as the important guiding significance to our research. We have studied these comments carefully and have made the correction which we hope meet with approval. The revised portion are marked with tracked changes in the revised manuscript.

Reviewer 2 Report

This study developed several functional (I think they are intra-locus) markers for five loci responsible for the differences in grain size related traits between Zhendao and R2027. Then, the authors developed derived lines from the cross between these two lines, genotyped them with the markers, and selected good restorer lines. The study in itself appeared to have a sound policy and method. However, I conclude that this manuscript is very hard to be accepted as an article with novel information and originality in an international journal, Agronomy. In addition, there are many parts which are immature expressions as an scientific article. I recommend the authors to submit this manuscript to other local journals after minor revisions. The reasons are as follows:

(1) The five loci and the polymorphic regions within them the authors detected and handled in this study were mostly the same as Zhang et al. (2020) (Rice 13: 63). Therefore, the authors simply adapted the already published information for markers to their own materials. There was no originality, at least, for marker development. The authors may claim that the sequences of primers are different from those of Zhang et al. (2020). However, of course, it can never claim the originality.

(2) The authors emphasized that the developed restorers with large grain size are useful for the breeding of hybrid rice cultivars. However, in hybrid cultivars, all of the loci which locate different alleles between the parents, are, of course, heterozygous. Therefore, in order to express superior effects on grain size (larger size in this case), both the restorer and male sterile lines (and also maintainer) should have the same superior alleles at most of the grain size loci. The authors never discussed on this issue. Alternatively, most of the superior alleles should be complete or over dominant over the counterpart alleles. However, this study only showed the effects of grain size alleles in the homozygous genotypes, not in the heterozygote, although the authors certainly grew the heterozygous F1 in the summer of 2018. If the authors are not deeply attached to the hybrid rice breeding, and delete all related parts from the manuscript, the above problems, at least, might be resolved, although it will make this study a very usual simple report, not an article.

Other minor point to be revised are as follows:

L5: Fix the font of the authors.

L20: genes => loci

L22: Were the lines RILs? Were they randomly selected from F2?

L25: Among the alleles from Zhendao, (?)

L26: …1000-grain weight, compared with R2027. gl7 and gs6

L28: these grain size genes => these grain size alleles

L29: dominant genes => dominant alleles

L31: “Gene” is too obvious term as a keyword.

L40: , and is an … (?)

L43:  …by multiple quantitative trait loci (QTLs), and many major loci and QTLs …

L44: genes => loci

L51: indica and japonica should be in italic.

L52: chromosome 5

L55: by decreasing the expression of

L62: chromosome 6

L63: by decreasing the number of

L66: The large grain allele, tgw6 of

L71: granular => grain size

L74: GW7 and GW8 should be in italic, if they express locus name.

L75: Japonica => japonica

L78: Xie et al.

L80: The grain length of the genotype involving these two large grain alleles was significantly increased compared with the genotypes with each allele of them, implying that there was an epistatic interaction effect between these two loci (for example).

L90 and others: Please follow the rule for description of the subsection in Agronomy. For example in L90, “2.1 Plant materials (?) (Italic, and the same font in the text)”

L99: Delete “amplified”.

L104: State the reference or web site for Primer Premier 6 software.

L109: enzyme

L115: I think 25 micro litter of polymerase premix is too much. This may be corrected.

L117: 2 x PCR Buffer (?)

L132: The calculation method for 1000-grain weight is too obvious, and might be omitted.

L144: “P” and “t” should be in italic.

L145: What is WT? Wild type? So, which is the wild type, Zhendao or R2027?

L147: In the L96-99, sequenced loci were five, not nine.

L148: difference => polymorphism

L148: genes =>loci

L149: these genes => other five loci, GW5, GS6, GW6a, TGW6, and GL7, and…

L149: How did the authors evaluate the expression of these loci? No description was found in the Methods.

L153 and others: State the references for the previous findings.

L159: How did the authors detect “cDNA” sequence of TGW6? No description was found in the Materials and Methods.

L169: Development of functional markers for five grain size genes (or alleles)

L191 (Table 1): TGW6a_SNP-F, R  In the text (L184), this is TGW6_SNP-F, R.

L193: a pollen donor (?)

L200 and Figure 3: F3, F7 => F3, F7

L202: genetic effects (?)

L211: Please state the procedure for the detection of restorer alleles with markers in the Materials and Methods.

L213: What is “high-quality”?

L223: What is “dominant grain weight”?

Table 2: What is “Rarerice”? Zhendao?

Table 2 and also Abstract: How did the authors assign the dominant (with capital letter, Gl7, Gw6a, Gs6, Gw5, Tgw6) and recessive alleles in this study?

L256: Zhang et al. showed functional markers for 14 grain size loci, and these loci included all of the five loci in this study. Therefore, this study provided no more “broadly” MAS.

L267, 268 and others: granule => grain

L269: developed

L278: Male sterile

L278: indica should be in italic.

L279: we

L285: What is “It exists alone”?

L285: we examined… but none of them were polymorphic.

L286-289: I can hardly understand these sentences.

L290: This Conclusion section did not involve any new information, and could be omitted.

Reference section: Please follow the rule of Agronomy to describe this section. For example, 1. Xing, Y.; Zhang, Q. Genetic and molecular bases of rice yield. Annu Rev Plant Biol 2010, 61, 421-442.

Author Response

Dear Editors and Reviewers:

Thank you for your professional reviewer comments and valuable suggestions on our manuscript entitled “Breeding of long-grain restorer of indica-japonica hybrid rice by using genetic effects of grain shape genes” (ID: 1972460). Those comments are of great help for revising and improving our manuscript, as well as the important guiding significance to our research. We have carefully studied all these comments and made corrections which we hope meet with approval. The revised portion are marked with tracked changes in the revised manuscript. The responses to the reviewer’s comments are as following:

Responses to the comments of Reviewer #2

This study developed several functional (I think they are intra-locus) markers for five loci responsible for the differences in grain size related traits between Zhendao and R2027. Then, the authors developed derived lines from the cross between these two lines, genotyped them with the markers, and selected good restorer lines. The study in itself appeared to have a sound policy and method. However, I conclude that this manuscript is very hard to be accepted as an article with novel information and originality in an international journal, Agronomy. In addition, there are many parts which are immature expressions as an scientific article. I recommend the authors to submit this manuscript to other local journals after minor revisions. The reasons are as follows:

Major comments:

  1. The five loci and the polymorphic regions within them the authors detected and handled in this study were mostly the same as Zhang et al. (2020) (Rice 13: 63). Therefore, the authors simply adapted the already published information for markers to their own materials. There was no originality, at least, for marker development. The authors may claim that the sequences of primers are different from those of Zhang et al. (2020). However, of course, it can never claim the originality.

Response: Thank you very much for your important suggestion. In this study, on the basis of Zhang et al.' research, the grain-shape genes in Zhendao and R2027 lines were detected, and the five loci and their polymorphisms were found out. These five loci and their polymorphic regions, and used the corresponding molecular markers of these five granule genes for detection. The detection results further verified the accuracy of previous studies. In this study, we want to use the aggregation of different grain shape genes between these two varieties to further improve the grain shape of rice, which can improve the breeding process of an improved line.

  1. The authors emphasized that the developed restorers with large grain size are useful for the breeding of hybrid rice cultivars. However, in hybrid cultivars, all of the loci which locate different alleles between the parents, are, of course, heterozygous. Therefore, in order to express superior effects on grain size (larger size in this case), both the restorer and male sterile lines (and also maintainer) should have the same superior alleles at most of the grain size loci. The authors never discussed on this issue. Alternatively, most of the superior alleles should be complete or over dominant over the counterpart alleles. However, this study only showed the effects of grain size alleles in the homozygous genotypes, not in the heterozygote, although the authors certainly grew the heterozygous F1 in the summer of 2018. If the authors are not deeply attached to the hybrid rice breeding, and delete all related parts from the manuscript, the above problems, at least, might be resolved, although it will make this study a very usual simple report, not an article.

Response: Thank you for your very original opinion and important suggestion. In fact, In fact, there are still some parts of this research that need to be further verified, such as the performance of F1 after the selected restorer lines are mated with different CMS or P/TGMS lines to further confirm their heterosis. The main idea of this paper is to show that through molecular marker assisted selection and aggregation of grain shape genes, restorers with long grain deformation and similar yield advantages can be bred, which will be further verified in the next step. Thank you again for your valuable comments.

Thank you for your comments and understanding on the correction of many grammatical errors and English proficiency in the manuscript. We are deeply sorry for this. On this basis, we made serious corrections, and found teachers and students to make serious corrections and improve the manuscript. All these changes are marked with tracked changes in the revised manuscript.

We appreciate for Editors/Reviewers’ warm work earnestly and hope that the correction will meet with approval.

Once again, thank you very much for your comments and suggestions. We are looking forward to hearing from you.

Best regards,

Corresponding author: Lianping Sun

Key Laboratory for Zhejiang Super Rice Research, Chinese National Center for Rice Improvement and Stat Key Laboratory of Rice Biology, China National Rice Research Institute, HangZhou, 311402, Chinam (G.H.)

E-mail: sunlianping@caas.cn

October 21, 2022

Reviewer 3 Report

The paper is well written and shows very interesting results which may considered supporting tools in the rice breeding.

Author Response

Response: Thank you very much for your careful review and important comments. We tried our best to improve the manuscript and made some changes in the manuscript. These changes will not influence the content and framework of the manuscript. We appreciate for Editors/Reviewers’ warm work earnestly and hope that the correction will meet with approval. Once again, thank you very much for your comments and suggestions. We are looking forward to hearing from you.

Reviewer 4 Report

The article titled in ''Breeding of long-grain restorer of indica-japonica hybrid rice by using genetic effects of grain shape genes'' is a good idea that serve rice production sustainability. The authors in this study, developed functional markers for five grain-size genes. Moreover, they detected the restorer genes Rf1 and Rf4 in the recombinant inbred lines. These finding can be good tool for designing a breeding program for improve grain size in three-line system hybrid rice.

  For the authors, please carefully check the English spelling and try to add more reference in the discussion part. 

Reviewer 5 Report

Title: too much „genetics“ – genetic effects of ...genes

In the key words: to increase probability of citation of the article wider scope of key words is required also some modifications of presently selected words - rice is replication of the title rice, international name of rice is missing, also Poaceae family might be mentioned, grain morphology, quantitative trait loci

Abstract – what is the best outcome among received

Introduction

Rows 31-34: the first sentences provide important arguments , although reference for it is missing.

The names of  cultivars  should be provided in unique style following international requirements,

presently in the abstract - variety Zhendao and in the row 72 - Japonica varieties

Methods

rows: 93-94   for „reported..“ reference is missing

Devices for sequencing are missing

references for software and other methodical approaches are missing

restriction enzyme producer is missing

Young rice leaves from the seedling stage – exact seedling age and morphology should be mentioned, also how selection of the leaves was performed

Row 125: „ The grains on the main panicle were harvested“ – all description about cultivation conditions is missing

Results

In all titles of illustrations the name of the plant is missing

Result chapter can not be finished by table and figure.

Discussions

No information where (in terms of soil and climate or geography) assessed cultivars are grown.

Results of the work should be discussed with reference to what others have done in the field. International context of the current discussion is missing and it is not relevant for the international journal. Reference list is very poor.

The table in supplement requires improvement concerning

1) title content – too brief

2) absence of explanation of abbreviation

3) till word is not found in the international dictionaries.

Round 2

Reviewer 2 Report

I cannot found the revisions against most of my comments. This revised manuscript was scarcely improved. My conclusion does not change. This study has no scientific originality, and should be rejected as an article in international journal, like Agronomy. 

Author Response

Response: Thank you for your very original opinion and important suggestion. In fact, there are still some parts of this research that need to be further verified, such as the performance of F1 after the selected restorer lines are mated with different CMS or P/TGMS lines to further confirm their heterosis. The main idea of this paper is to show that through molecular marker assisted selection and aggregation of grain shape genes, restorers with long grain deformation and similar yield advantages can be bred, which will be further verified in the next step. Thank you again for your valuable comments.